# Assessment of knowledge and perceptions towards diabetes mellitus and its associated factors among people in Debre Berhan town, northeast Ethiopia

**Wondimeneh Shibabaw Shiferaw**[1]***, Abel Gatew**[2]**, Getnet Afessa**[2]**, Tsedale Asebu**[1]**, Pammla Margaret Petrucka**[3]**, Yared Asmare Aynalem**[1]

**1** Department of Nursing, College of Health Sciences, Institute of Medicine and Health Science, Debre Berhan University, Debre Berhan, Ethiopia, **2** Clinical Nurse at Debre Berhan Referral Hospital, Debre Berhan, Ethiopia, **3** College of Nursing, University of Saskatchewan, Academics without Borders, Saskatoon, Canada

\* wshibabaw2015@gmail.com

**Data Availability Statement:** All relevant data are within the paper and it's supporting information files. There is no separate data set to share.

## Abstract

### Background

Globally, diabetes is a major public health burden that results in more than 3.2 million adult deaths per year. Currently, diabetes is increasingly becoming a major threat to global public health, particularly in Sub-Saharan Africa. Although previous studies emphasized knowledge and health beliefs about diabetes among patients living with diabetes, there is minimal evidence about knowledge and perception of risk for developing diabetes at the community level.

### Objective

This study aimed to assess the knowledge and perception of diabetes mellitus and its associated factors among people in Debre Berhan town, northeast Ethiopia.

### Methods

A community-based cross-sectional study was conducted among 423 participants. The study was carried out from 25 February to 10 March 2019. Data were collected using a structured pretested questionnaire through face-to-face interviews. Data were entered into Epi data V 3.1 and exported to SPSS V 24 for analysis. A variable with p< 0.2 in bivariable analysis was entered into multivariable logistic regression. During multivariate analysis, variables with a p value of $\leq$ 0.05 were considered significantly associated.

### Result

A total of 237 (56.02%) participants had good general knowledge about diabetes mellitus. In the multivariable analysis, participants who were single (AOR = 9.08, CI: 1.72–48), had a family history of diabetes (AOR = 2.83; CI: 1.10–7.24), and had exposure to health

**Funding:** The authors received no specific funding for this work.

**Competing interests:** The authors declare that they have no competing interests.

education (AOR = 3.27; CI: 2.02–5.31) were associated with good knowledge. In this study, few respondents (20.1%) felt that they had a higher risk of developing diabetes. Two-thirds of respondents (62.4%) saw diabetes is a serious disease. On the other hand, approximately 67% agreed to the perceived benefits of screening.

## Conclusion

Almost half of the Debre Berhan community was found to have inadequate knowledge of diabetes mellitus. Married, higher educational status, exposure to health education, and family history of diabetes mellitus were significantly associated with good knowledge. The perceived risk of developing diabetes was low at the community level, although many respondents felt that behavior change is important in the prevention of diabetes. Therefore, policy makers, healthcare managers, and healthcare workers need to work cooperatively to foster community knowledge towards diabetes mellitus.

## Background

Diabetes mellitus (DM) is a common and devastating chronic disease [1]. Worldwide, the burden of DM is rising dramatically and reaching epidemic proportions [2]. The World Health Organization (WHO) estimates that DM affects at least 285 million people and causes 3.2 million deaths, equating to 8700 deaths every day [3]. Due to sedentary lifestyles, rapidly growing urbanization, and modified diets, tripling of the magnitude of diabetes mellitus is predicted within the next 25 years, as well as an overall global burden escalation [4]. According to WHO estimations globally, there are approximately 422 million adults living with diabetes mellitus [5], while the International Diabetes Federation (IDF) estimates that 382 million people in 2013 will rise to 592 million by 2035 [6]. A more recent estimate suggests that the global prevalence will reach 642 million people in 2040 [7].

In Sub-Saharan Africa (SSA), more than 12 million people have diabetes, and there are 330,000 diabetes-related deaths, yet less than 1% of health expenditure is allocated for diabetes [8]. In Ethiopia, approximately 4.36% (1.9 million) of the population was likely to live with diabetes, and the number of deaths attributed to diabetes reached 34,262 in 2013 [6]. Ethiopia is among the top four countries for high adult diabetic populations in developing countries [9]. Overall, diabetes seriously increases the risk of costly problems, including emotional distress, heart attack, stroke, kidney damage, blindness, neural damage leading to amputation, and reduced quality of life [10].

Public knowledge about diabetes helps combat not only the disease itself but also its complications and medical and socioeconomic consequences [11]. Simple lifestyle modifications, such as a healthy diet that includes reducing sugar intake, are considered to be essential for the prevention and reduction of the incidence of diabetes mellitus [12, 13]. Numerous studies have revealed that the proportion of good knowledge regarding diabetes mellitus was 27% in Kenya [14], 49% in Debre Tabor town, Ethiopia [15], 52.5% in Bale Zone administrative, Ethiopia [16], 41.9% in Malaysia [17], 49.9% in India [18], 28.2% in rural Indian state [19], and 15% in Sudan [20]. Several studies undertaken in different countries have shown that age, gender, educational level, socioeconomic status, and family history of diabetes mellitus were associated with good knowledge about diabetes mellitus [15, 21–23].

This study is based on the Health Belief Model (HBM), which consists of perceived threat, perceived susceptibility, perceived severity, perceived benefits, perceived barriers and cues for

action [24]. The aim of this model is to increase the perception of individuals about a health threat and direct their behaviours towards health. Likewise, it focuses on a person's health-related behavior and belief in predicting future actions. The perceived risk of people developing diabetes mellitus is considered to be the primary motive to change within the Health Belief Model, which assumes that the higher the perceived threat, the more likely an individual will modify his or her behavior to circumvent that threat [25]. It may be an important motivating factor for preventative health behaviors and control of disease [26]. In addition, according to this model, the decision to participate in preventive and screening programs is determined by many factors, such as awareness of the impact of disease on their health (perceived severity), perceived benefits of undergoing preventive measures, and perceived barriers and costs of the screening methods [24, 27]. Therefore, understanding individuals' viewpoints and beliefs is essential for developing strategies to prevent, control, and increase awareness of the health risks of this disease [28].

Previous studies emphasized knowledge and attitudes towards diabetes mellitus. Local evidence is limited on public knowledge and risk perceptions towards DM using the health belief model in Ethiopia. Therefore, this study aimed to assess knowledge and risk perceptions towards diabetes mellitus and its associated factors among Debre Berhan community members. The results of this study may provide baseline data for future intervention programs to promote early detection and early management of diabetes mellitus.

## Methods

### Study design and setting

This community-based, cross-sectional study was conducted from 25 February to 10 March 2019 among Debre Berhan community members. Debre Berhan is one of the 13 zones of the Amhara regional state and the town of the North Shoa Zone. The town has nine kebeles, among which seven are primarily urban populations, while the remaining two kebeles have both urban and rural populations. The total population was 108,825, within 25,308 households. Moreover, there is one government hospital, one private hospital, three government health centers, nine health posts and 18 private clinics [29].

### Study participants

All adults living in Debre Berhan town for at least six months, aged greater than or equal to 18 years and consenting to participate were included in the study. Persons who were critically ill during the data collection period and had complete loss of hearing and known DM patients were excluded from the study.

### Sample size estimation

Sample size was calculated for each study objective. For the first objective, the sample size was determined based on the formula $(Z1-\alpha/2)2*(p(1-p))/d2$, where $(Z1-\alpha/2)$ = 1.96, p = the proportion with knowledge of diabetes mellitus, which is 52.5% from a previously published study [16], and d = 0.05. We added 10% to the calculated sample size to compensate for the anticipated non-response rate, bringing the total sample size to 418 community members. Similarly, for the second objective, appropriate sample size was estimated using a single proportion formula, where n is the need sample size; d, marginal error (d = 0.05); Z, the required degree of accuracy at 95% confidence level, which is 1.96; P = 0.5 (50%) level of risk perception towards diabetes, as there was no study conducted in the study area to the best of literature search made. Using the above formula, after adding a 10% non-response rate, the final sample size of

the study was 423 community members. Therefore, for this study, the largest sample size was taken to include 423 eligible study participants.

## Sampling procedure

The numbers of households were obtained from the community's administrative office. Three kebeles were selected by the lottery method out of nine kebeles in the town. Proportional allocation was used to obtain the desired number of samples from each kebele. Systematic random sampling was used to select the study unit among households. The first house was selected by the lottery method, and the next households were selected after calculating sampling intervals (K = 3). Then, every 3 households were taken until the required sample size was reached. The first person to be encountered in the household meeting the age criteria was interviewed. For those who failed to meet the inclusion criteria, a second person was interviewed, and if more than one individual meeting the age criteria was present in the same household, a lottery method was used. If there is no respondent in the selected households in two visits, the next household was considered until the desired sample size was achieved.

## Study variables

Knowledge and perception towards DM were the outcome variables. Sociodemographics, source of information, and family history of DM were considered independent covariates.

## Data collection tool and procedure

A data collection tool was developed from previous similar literature [19, 30, 31] that contains socio-demographic data, knowledge, and perceptions based on the Health Belief Model. Moreover, the three-part data collection tool was designed based on study objectives. The first part of the questionnaire focused on the sociodemographic information that included age, sex, marital status, level of education, occupation, average family monthly income, family history of diabetes mellitus, exposure to health education about DM, and source of health information. The second part assessed the level of knowledge about diabetes mellitus among the study participants and the level of participants' understanding of various aspects of DM, including definition, causes/risk factors, signs/symptoms, control and management. Respondents answered either "yes", or "no", or "I do not know" options. Correct responses were scored as one point, while the responses "No" or "I do not know" were scored as zero points. All respondents with below the mean score on knowledge questions were included among those with poor knowledge, while respondents scoring above the mean on knowledge questions were regarded as having good knowledge.

The third part of the tool is regarding perceptions towards DM based on the constructs of the Health Belief Model scale; this scale has five subscales: Perceived susceptibility to having disease was assessed by using three items. The first item inquired as to "My chances of getting diabetes mellitus in the next few years are high". The second regarding perceived seriousness of DM was assessed by four items (e.g., "If I had diabetes, I would be worried and depressed"). Perceived benefits of undergoing a preventive measure and screening were assessed by five items (e.g., "I believe maintain a normal body weight help to control diabetes."). Finally, perceived barriers to screening and healthy lifestyle were evaluated using five items (e.g., "I think having a regular health check-up takes too much time.") [24, 27]. All items of the subscales use five-point Likert-type response choices: Strongly Agree (5 points), agree (4 points), neutral (3 points), disagree (2 points) and strongly disagree (1 point). Using previous studies [32, 33], we categorized agree, neutral and disagree as a baseline. Each of the subscales was assessed separately, and the total score was not calculated. Subscale scores were calculated for each

construct. Higher scores indicated stronger feelings about that construct. In the original test, Cronbach's alpha coefficients for the five subscales were observed to fall between 0.65 and 0.87. In this study, Cronbach's alpha coefficients of 0.83 were observed for the five subscales. Data were collected through face-to-face interviews by six trained data collectors. The principal investigator of the study was controlling the overall activity.

## Data quality assurance

Pretesting was performed among 5% of study participants two weeks prior to the actual data collection period among non-sampled kebeles to avoid information contamination bias. The questionnaire was modified based on the responses to the pretest. In addition, the questionnaire was initially developed in English, then translated to the local language (Amharic) by an expert and then back to English to ensure consistency. Training on data collection was given to data collectors and supervisors for one day before actual data collection. All data collectors were trained on their responsibilities for the purposes of the study, how to collect the data, how to maintain confidentiality, and how to ensure genuine replies on questions. Furthermore, the principal investigator strictly followed the overall activities of the data collection on a daily basis to ensure the completeness of the questionnaires and to give further clarification. Double data entry was employed.

## Data processing and analysis

The collected data were checked for completeness, coded, entered into Epi Data VS 3.1 and then exported to SPSS VS 22 for analysis. Descriptive statistics were used to describe the study participants in relation to relevant variables. Then, bivariate and multivariate logistic regression was used to identify the possible associations between independent and outcome variables. Variables with p-values less than 0.2 during bivariable analysis were exported to multivariable analysis to control for confounding effects. Variables with a p-value of $\leq 0.05$ in the multivariable analysis were considered significantly associated with the outcome variable. Assumption tests and model goodness of fit were checked. Finally, the results of the study are presented in tables, figures, and text.

## Ethics consideration

This study was approved by the ethical review committee of the Institute of Medicine and College of Health Sciences, Debre Berhan University. An official letter of permission was provided to the Debre Berhan town administration. After explaining the purpose of the research to the study participants, verbal informed consent was obtained prior to data collection. The data obtained were maintained in a manner to ensure confidentiality.

## Results

### Sociodemographic characteristics of the study participants

A total of 423 respondents participated, yielding a response rate of 100%. Within this sample, 231 (54.6%) were females; nearly half of the participants [195 (46.1%)] were married. More than half [235 (55.6%)] of the study participants were educated at the college and above; nearly two-thirds (64.3%) had an average family income greater than 1,540 Ethiopian birrs per month. Most of the study participants, 255 (60.3%), were previously exposed to health education about DM (Table 1).

**Table 1. Sociodemographic characteristics of study participants in Debre Berhan town, northeast Ethiopia.**

| Variables | Category | Frequency | Percent (%) |
|---|---|---|---|
| Sex | Female | 231 | 54.6 |
| | Male | 192 | 45.4 |
| Marital status | Married | 195 | 46.1 |
| | Single | 194 | 45.9 |
| | Divorced/separated | 18 | 4.3 |
| | Widowed | 16 | 3.8 |
| Educational status | Unable to read and write | 28 | 6.6 |
| | Grade 1–4 | 25 | 5.9 |
| | Grade 5–8 | 52 | 12.3 |
| | Grade 9–12 | 83 | 19.6 |
| | College and above | 235 | 55.6 |
| Occupation | Housewife | 58 | 13.7 |
| | Student | 86 | 20.3 |
| | Merchant | 70 | 16.5 |
| | Government/private employee | 167 | 39.1 |
| | Day laborer | 42 | 9.9 |
| Average monthly income | ≤1,539 ETB | 151 | 35.7 |
| | ≥1540 ETB | 272 | 64.3 |
| Exposure to health education about DM | Yes | 255 | 60.3 |
| | No | 168 | 39.7 |
| Family history of DM | Yes | 54 | 12.8 |
| | No | 369 | 87.2 |
| Source of information | Public media | 121 | 47.5 |
| | Health care worker | 79 | 31.0 |
| | Friends | 34 | 13.3 |
| | Other (teacher & religious leader) | 21 | 8.2 |

N.B. ETB; Ethiopian birr.

## Knowledge of participants about diabetes mellitus

In general, the total mean score for correctly answered knowledge questions was 11.4 ±4.9. Two hundred thirty-seven (56.02%) scored above the mean and were considered to have good knowledge. In contrast, 186 (43.98%) scored below the mean were poorly knowledgeable. Among 423 participants, approximately 47% responded correctly defined DM as high levels of sugar in the blood, and approximately 40.7% said DM is incurable. In addition, regarding risk factors, the percentage of participants correctly naming family history of DM was 54.4%; being overweight or obese was 64.1%; and sedentary lifestyle was 60.8% of participants. Significant proportions of study participants (79.7% and 79.2%) reported excessive hunger and feeling of weakness as signs and symptoms of DM, respectively. Moreover, related to the management of DM, they described DM as being controlled by insulin injection (77.1%), regular exercise (66.2%) and a healthy diet (73.8%) (Table 2).

## Factors associated with participants' knowledge about DM

In the multivariable analysis, participants who were married were almost eleven times more likely to be knowledgeable than those who were widowed (AOR = 10.97, CI; 2.33, 51.64). Likewise, participants who were unable to read and write were 81% less likely to have good knowledge than those who had college and above educational status (AOR = 0.19, CI; 0.53, 0.68).

**Table 2. Participants' knowledge about diabetes mellitus in Debre Berhan town, northeast Ethiopia.**

| Variables | Yes | | No | |
|---|---|---|---|---|
| | number | % | number | % |
| **What is/are DM** | | | | |
| DM is a condition of insufficient insulin production | 120 | 28.4 | 303 | 71.6 |
| DM is a condition of the body which not responding for insulin | 115 | 27.2 | 308 | 72.8 |
| DM is a condition of high level of sugar in the blood | 199 | 47.0 | 224 | 53.0 |
| DM is not curable | 251 | 59.3 | 172 | 40.7 |
| **What are the risk factors of DM** | | | | |
| Older age | 213 | 50.4 | 210 | 49.6 |
| Genetic or family history of diabetes mellitus | 230 | 54.4 | 193 | 45.6 |
| Being overweight /Obesity | 271 | 64.1 | 152 | 35.9 |
| Sedentary life /Poor dietary habits | 257 | 60.8 | 166 | 39.2 |
| **What are the signs and symptoms of DM** | | | | |
| Frequent urination | 206 | 48.7 | 217 | 51.3 |
| Excessive thirst | 214 | 50.6 | 209 | 49.9 |
| Excessive hunger | 337 | 79.7 | 86 | 20.3 |
| Weight loss | 223 | 52.7 | 200 | 47.3 |
| High blood sugar | 258 | 61.0 | 165 | 39.0 |
| Blurred vision | 233 | 55.1 | 190 | 44.9 |
| Slow healing of cuts and wounds | 249 | 58.9 | 174 | 41.1 |
| Feeling of weakness | 335 | 79.2 | 88 | 20.8 |
| **Control and management of DM** | | | | |
| Insulin injection is available for control and management of DM | 326 | 77.1 | 97 | 2.9 |
| Tablets & capsule are available for control and management of DM | 226 | 53.4 | 197 | 46.6 |
| Regular exercise | 280 | 66.2 | 143 | 33.2 |
| Practices healthy diet | 312 | 73.8 | 111 | 26.2 |

Similarly, participants who had a family history of DM were almost three times more knowledgeable than those who did not have a family history of DM (AOR = 2.83, CI: 1.11, 7.25). Participants who had diabetes health education exposure were slightly more than three times as knowledgeable as those who did not have diabetes health education exposure (AOR = 3.28, CI; 2.02, 5.31) (Table 3).

### Perceptions towards diabetes mellitus based on the health belief model

The findings showed that 20.1% of the participants considered themselves to be at risk of developing DM in the future (perceived susceptibility). Additionally, 62.4% of the participants agreed to the statements related to the seriousness of the disease. Regarding the perceived benefits of the screening and preventive measure, approximately two-thirds of the participants believed that undergoing regular health care visits can help find DM early and save lives. Twenty-seven percent of the participants had experienced barriers to undergoing screening and implementing lifestyle changes, while 31.2% of participants never heard or read anything encouraging them to have regular health check-up and 25.1% of participants felt having a regular health check-up takes too much time (Table 4).

## Discussion

The current study showed that approximately 56.02% (95% CI: 53.82, 59.47) of the study participants were knowledgeable about DM. This result is in line with a study performed in the

**Table 3. Bivariate and multivariate logistic regression analysis to characterize factors associated with knowledge of diabetes mellitus in Debre Berhan town, northeast Ethiopia.**

| Variable | Category | Knowledge | | COR(95%CI) | AOR(95% CI) |
|---|---|---|---|---|---|
| | | Good Knowledge | Poor knowledge | | |
| Sex | Male | 130 | 62 | 1.05(0.69, 1.57) | 0.86(0.49, 1.49) |
| | Female | 154 | 77 | 1 | 1 |
| Marital status | Single | 129 | 65 | 8.6(2.36, 31.25)* | 9.08(1.72, 48.00)* |
| | Married | 144 | 51 | 12.24(3.35, 44.68)* | 10.97(2.33, 51.64)* |
| | Divorced | 8 | 10 | 3.467(0.72, 16.53) | 4.23(0.67, 26.58) |
| | Widowed | 3 | 13 | 1 | 1 |
| Educational status | Unable to read &write | 9 | 19 | 0.17(0.07, 0.39)* | 0.19(0.53, 0.68)* |
| | Grade 1–4 | 11 | 14 | 0.28(0.12, 0.65)* | 0.24(0.07, 0.82)* |
| | Grade 5–8 | 37 | 15 | 0.88(0.45, 1.72) | 0.87(0.36, 2.07) |
| | Grade 9–12 | 54 | 29 | 0.66(0.39, 1.14) | 0.72(0.37, 1.39) |
| | College and above | 173 | 62 | 1 | 1 |
| Occupational status | House wife | 34 | 24 | 0.88(0.26, 3.04) | 0.75(0.15, 3.65) |
| | Student | 56 | 30 | 1.16(0.35, 3.88) | 0.40(0.09, 1.77) |
| | Merchant | 44 | 26 | 1.05(0.31, 3.57) | 0.46(0.11, 1.95) |
| | Farmer | 9 | 4 | 1.40(0.27, 0.71) | 2.28(0.30, 17.10) |
| | Government employer | 125 | 42 | 1.86(0.57, 5.99) | 0.67(0.17, 2.64) |
| | Daily labor | 8 | 8 | 0.62(0.14, 2.76) | 0.57(0.09, 3.36) |
| | Driver | 8 | 5 | 1 | 1 |
| Exposure to health education | Yes | 202 | 53 | 3.99(2.60, 6.13)* | 3.27(2.02, 5.31)* |
| | No | 82 | 86 | 1 | 1 |
| Family history of DM | Yes | 44 | 10 | 2.37(1.15, 4.85)* | 2.83(1.10, 7.25)* |
| | No | 240 | 129 | 1 | 1 |

* For statistically significant variables at p<0.05 both at bivariable and multivariable analysis: 1 reference variable; COR: crude odds ratio; AOR; adjusted odds ratio.

Bale zone, Ethiopia, which reported 52.2% [16]. Our finding was higher than a study in Debre Tabor town at 49% [15], 41.9% in Malaysia [17], 15% in Sudan [20], 27.2% in Kenya [14], 49.9% in India [18], and 49.7% in Namibia [34]. This variation might be due to the study period since the studies were conducted, for instance, in Sudan (2014), Kenya (2010), India (2008), and Malaysia (2014). The other possible explanation could be that diabetes is now emerging as an epidemic of the 21st century. To curb this scourge of diabetes, studies have shown that knowledge is the greatest weapon in the fight against DM [35, 36]. In line with this evidence, the Ethiopia government has designed a strategy to increase community awareness and knowledge of diabetes mellitus through public media and incorporated noncommunicable disease prevention and control programs as one core component of health extension programs [37].

In the present study, approximately 40.5% of study participants knew the definition of DM, with 60.7% having good knowledge about symptoms of DM, 57.4% of participants correctly identifying the risk factors of DM, and 65.7% showing good knowledge on control and management of DM. This finding is in line with a study undertaken in a semi-urban Omani population on diabetes definition and classical symptoms, which were reported as 46.5% and 57.0%, respectively [38]. Likewise, a study conducted in Debre Tabor Town found that 60.3% knew the definition, 39.1% had good knowledge about symptoms of DM, 49% identified the risk factors of DM, and 44% respondents had good knowledge on the control and management of DM [15]. In contrast, our results were higher than those of a study conducted in Kenya on

**Table 4. Health belief model scores of participants in Debre Berhan town, northeast Ethiopia.**

| Variables | Category | Agree | Neutral | Disagree |
|---|---|---|---|---|
| | | N (%) | N (%) | N (%) |
| Perceived susceptibility | My chances of getting diabetes in next few years is great | 79(20.2) | 46(11.7) | 266(68.1) |
| | I feel I will get diabetes sometime during my life | 70(16.5) | 68(16.0) | 286(64.5) |
| | I believe all population are equally likely to develop diabetes | 106(24.3) | 45(10.3) | 285(65.4) |
| | Average score (%) | 85(20.1) | 53(12.5) | 285(67.4) |
| Perceived severity | If I had diabetes, I would be worried and depressed | 299(73.6) | 6(1.6) | 101(24.8) |
| | If I had diabetes, I would have to have my diabetes taken off by medication | 356(83.6) | 3(0.7) | 67(15.7) |
| | DM can be a serious disease if you don't prevent it. | 116(28.6) | 38(9.3) | 252(62.1) |
| | If I had diabetes, it would cause me to die | 285(66.1) | 2295.1) | 124(28.8) |
| | Average score (%) | 264(62.4) | 23(5.4) | 136(32.2) |
| Perceived benefits | I believe diabetes can be cured easily | 160(37.8) | 51(12.1) | 212(50.1) |
| | I believe maintain a normal body weight help to control diabetes | 275(64.6) | 57(13.4) | 94(22.0) |
| | Regular health care visit can help finding diabetes early and save my life | 373(88.2) | 18(4.3) | 32(7.5) |
| | I believe that eat low sugar snacks & low-fat meals prevent DM in the future | 295(77.0) | 31(8.1) | 57(14.9) |
| | I believe that regularly physical exercise will help to prevent diabetes | 298(70.5) | 64(15.1) | 61(14.4) |
| | Average score (%) | 284(67.1) | 44(10.5) | 95(22.4) |
| Perceived barriers | I don't want to know if i have diabetes or not | 72(16.8) | 128(30.0) | 227(53.2) |
| | I think having a regular health check-up takes too much time. | 106(32.9) | 44(13.7) | 172(53.4) |
| | Not having enough money would keep me from having a check-up. | 92(21.5) | 84(19.7) | 251(58.8) |
| | I never heard or read anything encouraging having regular health check-up. | 132(30.9) | 77(18.0) | 218(51.1) |
| | I could not have enough time to exercise | 85(19.8) | 18(4.2) | 325(76.0) |
| | Average score (%) | 114(27.0) | 70(16.6) | 239(56.4) |

DM sign/symptom (29%) and risk factors (26%) [14] and one in Nigeria on knowledge of respondents on risk factors of DM was 19.0% and symptoms of diabetes mellitus was 18.2% [39]. The differences might be due to the Ethiopia government's strategy to reduce the burden of noncommunicable diseases, particularly diabetes, through incorporating a noncommunicable disease prevention program under the health extension package, which includes extensive print media [40].

Study participants who were unable to read and write had 81% less likely good knowledge about DM than those whose educational status was college and above. These findings were supported by studies conducted in semi-urban Omani populations [38], in Debre Tabor [15], and in Bangkok [23]. This finding might reflect that respondents who had higher education would have the chance to obtain different information, such as leaflets and manuals, which make them more aware about DM. Furthermore, this more highly educated group is more likely able to communicate more easily with health care providers regarding questions or concerns.

In the present study, those who had diabetes health education exposure were almost three times more knowledgeable than those who did not have diabetes health education exposure. This finding was consistent with the study conducted in Bangladesh [21] and the finding also supported by the study done in Bale zone [16]. On the other hand, in the current study, participants who had a family history of DM were nearly three times more knowledgeable than those who did not have a family history of DM, which is similar to findings in a semi-urban Omani population [38], in Debre Tabor town [15], and in Southwest Ethiopia [41]. In this study, married individuals were eleven times and single individuals nine times more knowledgeable than those who were widowed, reflecting findings from a previous study performed in Southwest

Ethiopia [41]. This could be due to the proportion of participants who were married in our study sample. Therefore, the authors suggest that future study is required to investigate the correlation between marital status and level of knowledge.

According to the present study, approximately 20.1% of participants had perceived susceptibility to developing diabetes in the future. This is consistent with the findings in Rwanda [42] found that the majority of respondents thought that there was almost no chance or no chance at all that they would develop diabetes. Additionally, 62.4% of the participants correctly agreed to the statements related to the seriousness of the disease. Regarding perceived severity, our finding is lower than a study done in Rwanda, where approximately 79.4% thought of DM as a severe disease [43] and 71% in Namibia reported as a serious disease [34]. This may reflect that there are other diseases that take priority and are viewed as more serious. In addition, not knowing the seriousness of a condition could be due to a lack of education interventions that can assist in alerting communities about these health conditions.

Concerning perceived benefits of the screening and preventive measure, approximately 67% of the participants believed that undergoing regular health care visits can help find DM early and save lives. Twenty-seven percent of the participants had experienced barriers to undergoing screening and implementing lifestyle change. For example, 31.2% of participants never heard or read anything encouraging about having regular health check-up; 25.05% of participants considered having a regular health check-up takes too much time; and 21.7% of the participant did not have enough money, which would keep them from having a check-up. This finding suggests that there is a need for education campaigns in the district to reduce barriers to diabetes screening about all aspects of barriers, such as time consumption and misinformation. It is therefore important to identify interventions that reduce people's perceived barriers despite their levels of knowledge of diabetes mellitus.

## Limitation of the study

Despite extensive efforts, the results of this study are subject to certain limitations. First, the cross-sectional nature of the study design makes it impossible to form causal relationships between exposure and outcome variables. Second, the study is conducted in one community and one cultural group; therefore, the findings might not necessarily be generalized to be the same in rural areas.

## Conclusion

This community-based cross-sectional study showed that the overall knowledge about DM was moderate. Educational status, marital status, family history of DM and exposure to health education had significant associations with the mean knowledge of the study participants. The majority of the sample had a low perception of the risk for developing diabetes. Therefore, policy makers, healthcare managers, and healthcare workers should plan health education interventions that can enhance public knowledge and the correct perception of the risk of developing diabetes. The use of the HBM, as in this study, could form the framework for further research on diabetes prevention among the Ethiopian population. Furthermore, future nationwide studies based on this framework are recommended to help develop more effective DM prevention interventions among the adult population in Ethiopia.

## Supporting information

**S1 File. English version of the questionnaire.**
(DOCX)

**S2 File. Amharic version of the questionnaire.**
(DOCX)

## Acknowledgments

The authors express their appreciation to Debre Berhan University and Debre Berhan town administration for their kind cooperation during data collection. The authors are grateful to the study participants and data collectors.

## Author Contributions

**Conceptualization:** Wondimeneh Shibabaw Shiferaw.

**Data curation:** Yared Asmare Aynalem.

**Formal analysis:** Abel Gatew, Getnet Afessa, Tsedale Asebu, Yared Asmare Aynalem.

**Investigation:** Getnet Afessa, Tsedale Asebu, Yared Asmare Aynalem.

**Methodology:** Wondimeneh Shibabaw Shiferaw, Abel Gatew, Getnet Afessa, Tsedale Asebu, Pammla Margaret Petrucka, Yared Asmare Aynalem.

**Software:** Wondimeneh Shibabaw Shiferaw, Pammla Margaret Petrucka.

**Supervision:** Abel Gatew, Getnet Afessa, Tsedale Asebu, Yared Asmare Aynalem.

**Validation:** Abel Gatew.

**Visualization:** Getnet Afessa.

**Writing – original draft:** Wondimeneh Shibabaw Shiferaw, Pammla Margaret Petrucka, Yared Asmare Aynalem.

**Writing – review & editing:** Wondimeneh Shibabaw Shiferaw, Pammla Margaret Petrucka.

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
