## [Decision Letter · Decision Letter 0]

22 Jun 2020

PONE-D-20-04491

Assessment of Knowledge and Perception towards Diabetes Mellitus and Its Associated Factors among People in Debre Berhan Town, Northeast Ethiopia:

PLOS ONE

Dear Dr. shiferaw,

Thank you for submitting your manuscript to PLOS ONE. After careful consideration, we feel that it has merit but does not fully meet PLOS ONE’s publication criteria as it currently stands. Therefore, we invite you to submit a revised version of the manuscript that addresses the points raised during the review process.

We look forward to receiving your revised manuscript.

Kind regards,

Cindy Gray, Ph.D.

Academic Editor

PLOS ONE

Additional Editor Comments:

Abstract:

I am not clear what an institutional-based study is. Also, the end of the 3rd paragraph in the introduction suggests there have been a number of community studies including two in Ethiopia.

This sentence is not clear to me: The finding shows that 20.1% of the participants were agreed related to perceived susceptibility of the disease.

This is not very informative without some more context: About 62.4% of the participants agreed to the statements related to the seriousness of the disease.

Please keep decimal points the same.

The conclusion needs to go further than simply more or less restate the results.

Where is the HBM here?

Introduction:

Keep statements about increasing global prevalence in the same place (1st paragraph) and perhaps keep developing countries statements together.

Keep the Health Belief model statements together

Scantly is not a very scientific term.

The link between the HBM and its utility for interventions needs to be more clearly made.

All in all the introduction needs some careful reworking to improve the coherence and logic of the argument made.

Method:

It is not entirely clear what variable your sample size calculation is based on

The sampling procedure is very hard to follow. It needs carefully re-written taking account of the comments below

Lottery method was used to select the study unit among households who have more than one eligible participant. I don’t understand what you mean here.

I also don’t understand this next households were selected after calculating skip fraction (K=x)

There appears to be some repetition, which makes the sampling procedure hard to follow.

In Study variables (and elsewhere), is According to the current study needed?

Data collection tool – How can individuals give information about community-level variables

I don’t understand this: According to this study, good knowledge reflected when respondents scored above the mean score on knowledge questions; whereas, poor knowledge was determined when respondents’ answers were below the mean score on knowledge questions. Is this a mean split?

Need references for the previous studies for the HBM scale categorisation

Additional ldata were collected through face to face interviews by six traineddata collectors – I don’t understand this

What modifications were done following the pre-testing?

Results:

Table 1 is hard to follow which categories belong to which variables

In the knowledge table – should it be clearer which answers are correct?

Sort out labelling of tables – there are mistakes. Figure 1 doesn’t add much – consider removing.

Why do you not highlight low education association with poor knowledge in the text?

Table 3 Title could be more informative

Table 4 needs percentages throughout – not just for average score – also what is the average score relating to? Is this numbers?

Should screening and healthy lifestyle be part of the same construct?

Again make sure decimal points for figures are consistent throughout

Discussion

What evidence is there for improvements in paragraph 1 – this argument needs to be made more compelling. – e.g. were the other studies done some time ago, are health systems etc better in Ethiopia?

Provide a reference for the Ethiopian Government’s NCD strategy in para 2

The consistency might be due to the widowed individuals are at risk of psychological distress or may be older?. This is too vague – what is the evidence for this?

The second limitation reads like a bullet point

All in all the discussion needs to more directly address the HBM and how the results might inform what interventions are needed.

In general the manuscript needs to be carefully checked for typos. There are a lot that need addressed.

Journal Requirements:

2. Please include additional information regarding the survey or questionnaire used in the study and ensure that you have provided sufficient details that others could replicate the analyses. For instance, if you developed a questionnaire as part of this study and it is not under a copyright more restrictive than CC-BY, please include a copy, in both the original language and English, as Supporting Information. Moreover, please include more details on how the questionnaire was pre-tested, and whether it was validated.

"none"

5. Please include a copy of Table 2 which you refer to in your text on page 11.

Reviewers' comments:

Reviewer's Responses to Questions

**Comments to the Author**

1. Is the manuscript technically sound, and do the data support the conclusions?

Reviewer #1: Yes

2. Has the statistical analysis been performed appropriately and rigorously? 

Reviewer #1: I Don't Know

3. Have the authors made all data underlying the findings in their manuscript fully available?

Reviewer #1: No

4. Is the manuscript presented in an intelligible fashion and written in standard English?

Reviewer #1: No

5. Review Comments to the Author

Reviewer #1: General points:

1. The authors have not provided the ethical statement requested by the journal

2. The authors need to use one of the data availability statements provided by the journal

3. The limitations section appears incomplete and contains what appears to be drafting notes

4. Insufficient care to distinguish types of DM – it is not a single condition

Introduction

• Change “which means” to “equating to” when describing deaths per day.

• Remove erroneous ‘g’ and rephrase sentence – “Numerous studies have revealed the proportion of good knowledge respecting g diabetes mellitus was reported as…”

• I would like to see the authors engage critically with the health belief model and justify its use.

• Use “scant” not “scanty”

Methods

• Check the grammar and punctuation carefully – there are missing full stops, shifts of tense and duplicated words eg:

o “The town of North Shoa Zone The town”

o “The total population were 108,825, within 25,308 households” [should be ‘was’]

o “All adults living in Debre Berhan town for at least for six months…”

• The authors should share the questionnaire used as an appendix or supplementary file.

• This sentence needs to be more precise and should be referenced: “Using previous studies, as a baseline we have categorized in to agree, neutral and disagree”

Results

• In table 1, income currency should be specified

• General checking of tenses required e.g. “was widowed” should be “were widowed”

• Table reporting knowledge of DM is labelled table 1, but it is the second table in the manuscript. The formatting of the table is also hard to read- we need additional spaces and or bold text between the sections relating to description of DM, risk factors, symptoms, and control

Discussion

• The discussion compares findings of the study to other relevant studies well, but it does not consider the meaning of the findings presented in the paper- what does this new study add to our knowledge of diabetes in this setting?

• Also, what do the findings suggest should happen or be prioritised? E.g. for those patients who experienced barriers to diabetes screening, what can/should be done?

• The final paragraph of the discussion is quite confusing and is not clearly written.

• The limitations paragraph is incomplete and contains drafting notes.

6. PLOS authors have the option to publish the peer review history of their article (what does this mean?). If published, this will include your full peer review and any attached files.

Reviewer #1: No

---

## [Author Response · Author response to Decision Letter 0]

20 Jul 2020

Author’s response to editor and reviewers 

Title: Assessment of Knowledge and Perception towards Diabetes Mellitus and Its Associated Factors among People in Debre Berhan Town, Northeast Ethiopia. 

Version 1: June 30, 2020

Dear editor 

To PLOS ONE, we are pleased to resubmit for publication version of “Assessment of Knowledge and Perception towards Diabetes Mellitus and Its Associated Factors among People in Debre Berhan Town, Northeast Ethiopia” with manuscript ID of “PONE-D-20-04491” for a review as original research in PLOS ONE. The comments of the editor and reviewers were highly insightful and enabled us to greatly improve the quality of our manuscript. Therefore, based on the editor and reviewers’ concerns we have made extensive edition in our manuscript. The formatting of the text and document (text sizes and grammatical errors) were also edited. In the following pages, we have addressed yours’ concerns in a point by point format. 

 We look forward to hearing from you at your earliest convenience. 

Once again, thank you for considering our manuscript in PLOS ONE! 

Kind regards,

Wondimeneh Shibabaw shiferaw 

On behalf of all the authors 

Author’s response to Editor Comments

Abstract:

1. I am not clear what an institutional-based study is. Also, the end of the 3rd paragraph in the introduction suggests there have been a number of community studies including two in Ethiopia.

Response: We thanks and accept your feedback. Based on your insightful feedback authors have been made appropriate modification in the revised manuscript. In fact you are right, two studies have been conducted in Ethiopia, however, both of them focus on knowledge, attitude, and practices towards diabetes mellitus. But, our study is unique because it emphasized on public knowledge and risk perception towards DM using the health belief model in northeast Ethiopia.

2. This sentence is not clear to me: The finding shows that 20.1% of the participants were agreed related to perceived susceptibility of the disease.

Response: we accept your concern and appropriate correction has been made in the revised manuscript. Therefore, the statement has been restated as “In this study, few respondents 20.1% felt that they have higher risk of developing diabetes”. 

3. This is not very informative without some more context: About 62.4% of the participants agreed to the statements related to the seriousness of the disease.

Response: we accept your comments and necessary correction has been made within the revised manuscript by rephrasing as “In the current study, two third of respondents 62.4% saw diabetes as a seriousness disease”. 

4. Please keep decimal points the same.

Response: we accept your feedback.

5. The conclusion needs to go further than simply more or less restate the results. 

Response: right you are. Based on your comments authors rephrase the conclusion. 

6. Where is the HBM here?

Response: in the revised document authors made further attempt to summarize the five pillars construct of HBM with short sentence particularly risk perception as “the perceived risk of developing diabetes was low at the community level, though many respondents felt that behavior change has importance in the prevention of diabetes’. 

Introduction:

1. Keep statements about increasing global prevalence in the same place (1st paragraph) and perhaps keep developing countries statements together. Keep the Health Belief model statements together

Response: We accept your suggestion and necessary correction has carried out throughout the introduction section. 

2. Scantly is not a very scientific term.

Response: we accept and correction has been taken. 

3. The link between the HBM and its utility for interventions needs to be more clearly made.

Response: To the best of our maximum effort the link between HBM and its usefulness for intervention has been clearly stated in the revised document page#4. Additionally, to make it clear the health belief model was developed to explain preventive health behaviours and control of disease. The aim of this model is increasing the perception of individuals about a health threat and directs their behaviours towards health. This model is a comprehensive pattern which represents the association between beliefs and behaviours; it has been widely applied to various health behaviours, especially screening behaviours and plays a significant role in prevention of the disease. Therefore, understanding individuals’ view points and beliefs is essential for developing the strategies of controlling diabetes. Moreover, the Health Belief Model is one of the models that widely used as a guiding framework for health behaviour interventions. 

4. All in all the introduction needs some careful reworking to improve the coherence and logic of the argument made.

5. Response: we accept your comments and extensive edition has been taken to the entire part of introduction section. 

 Method:

1. It is not entirely clear what variable your sample size calculation is based on

Response: The sample size was calculated based our study objective. The primary objective is to assess level of knowledge towards diabetes among Debre Berhan town residents northeast Ethiopia. Second, to determine the perceived risk of developing diabetes among Debre Berhan town community. Therefore, for the first objective sample size was calculated using single population proportion formula with an assumption of 95% confidence level, 5% degree of precision and 52.5% of respondents had good knowledge towards diabetes in accordance with previous literature report. For the second objective due to lack of available data on risk perception towards diabetes 50% proportion has been taken to estimate the sample size. Finally, by comparing the estimated sample size in both objectives, authors have decided to take the largest sample size which is found from the second objective (risk perception) which is 423. Additionally, to make it clear calculated the sample size of the present study the following assumption has been taken in to consideration, single population proportion was 50%, Za/2=95%, and 5% margin of error. 

2. The sampling procedure is very hard to follow. It needs carefully re-written taking account of the comments below

2.1. Lottery method was used to select the study unit among households who have more than one eligible participant. I don’t understand what you mean here.

Response: We accept your valuable feedback. This statement is our technical error and after evaluating we remove the sentence from the revised document. 

2.2. I also don’t understand this next households were selected after calculating skip fraction (K=x).

Response: right you are. It has been vague in the document, but now based on your comment necessary correction has been carried out. To make it clear, in the present study, the study unit was selected using systematic random sampling technique. First, the total frame size (number of households) and sample size was determined. Then we determine the number of sampling intervals K by dividing households by sample size. Then we select the random start (the first sampling unit) from the first interval in the frame with lottery method. Then every Kth unit was taken until the required sample size was achieved. Therefore, in this study K=N/n; 1,347/423=3.2 approximately 3. 

2.3. There appears to be some repetition, which makes the sampling procedure hard to follow.

Response: We thanks and appropriate modification has been made.

3. In Study variables (and elsewhere), is According to the current study needed?

Response: right you are and correction has been carried out.

4. Data collection tool – How can individuals give information about community-level variables

Response: In fact our narration has the problem. To make it clear, it is obvious that each individual is the part of the community, therefore by incorporating adequate sample of participants through scientifically sounded method, authors can inference appropriate conclusion based on his/her hypothesis. However, in the present study authors provide corrections with respect to the nature of variables are made to be considered as personal level. 

5. I don’t understand this: According to this study, good knowledge reflected when respondents scored above the mean score on knowledge questions; whereas, poor knowledge was determined when respondents’ answers were below the mean score on knowledge questions. Is this a mean split?

Response: This mean is not split, but to make it clear the measurement for knowledge about diabetes mellitus comprised of questions entailing the causes /risk factors, signs and symptoms and complications of diabetes mellitus. Using mean scores, the respondents were categorised into varying levels of knowledge namely 'good' and 'poor'. All respondents with below the mean score on knowledge questions were included among those with poor knowledge while respondents scoring above the mean on knowledge questions were regarded as having good knowledge. Therefore, the above statement has been provided appropriate correction within the revised manuscript. 

6. Need references for the previous studies for the HBM scale categorisation

Response: We accept your comments and necessary citation has been included in the revised manuscript. 

7. Additional l data were collected through face to face interviews by six trained data collectors – I don’t understand this

Response: right you are the sentence has the problem. Therefore, based on your insightful comments, modification has been carried out in the revised document as “Data were collected through face to face interviews by six trained data collectors’. 

8. What modifications were done following the pre-testing?

Response: after pre-test necessary correction has made in the document such as improved language clarity, arrangement of data from low to difficult questions, and modify the context of some questions. 

 Results:

1. Table 1 is hard to follow which categories belong to which variables

Response: we thanks so much. Based on your comment, authors change the design of the table to make clear which category is belong to which variables in the revised manuscript page#8-9 table 1. 

2. In the knowledge table – should it be clearer which answers are correct?

Response: your concern is very interesting. In the knowledge table 2 the authors state the each question items based on the response of the study participants either the question has negative or positive statement. However, when we calculating the overall mean score of knowledge questions, we have rearrange positive and negative statement based on the nature of the question. Hence, it has negative impact when estimating the overall mean score. 

3. Sort out labelling of tables – there are mistakes. 

Response: We accept your suggestion and correction has been taken for each. 

4. Figure 1 doesn’t add much – consider removing.

Response: We thanks so much. Based on your suggestion figure 1 had been removed from the document.

5. Why do you not highlight low education association with poor knowledge in the text?

Response: right you are this is our problem. But, now based on your comments authors clearly describe the association of educational status on knowledge of diabetes mellitus within the revised document as “participants who are unable to read and write were 81% less likely to have good knowledgeable than those who have college and above educational status (AOR=0.19, CI; 0.53, 0.68)”. 

6. Table 3 Title could be more informative

Response: we accept your suggestion and the title has corrected as “Bivariate and multivariate logistic regression analysis to characterize factors associated with knowledge of diabetes mellitus in Debre Berhan town, Northeast Ethiopia”.

7. Table 4 needs percentages throughout – not just for average score – also what is the average score relating to? Is this numbers?

Response: we accept your suggestion and apply percentage throughout the table 4. The average score is related to the response of participants (numbers) from each constructs of HBM. Subscale average scores were calculated for each constructs. Therefore, higher scores indicated stronger feelings about that construct. Hence, all HBM constructs are not able to sum up due to their different items within each constructs. Previous studies evidence also supports the present analysis. 

8. Should screening and healthy lifestyle be part of the same construct?

Response: By taking in to consideration your idea, authors extensive discuss on this issue then after decide to put HBM construct independently related to screening and healthy lifestyle. Therefore, we have restated this construct as “perceived barriers”. 

9. Again make sure decimal points for figures are consistent throughout

Response: We accept your suggestion and uniform decimal points have been used throughout the revised manuscript. 

Discussion

1. What evidence is there for improvements in paragraph 1 – this argument needs to be made more compelling. – e.g. were the other studies done some time ago, are health systems etc better in Ethiopia?

Response: To make it clear the possible variation could be the study period since the studies were conducted for instance in Sudan (2014), Kenya (2010), India (2008), and Malaysia (2014); because currently diabetes mellitus (DM) has become an increasing trend globally and a growing public burden related to this public knowledge and awareness have grown parallel. The other possible explanation could be diabetes is now emerging as an epidemic of the 21st Century. To curb this scourge of diabetes, it is known that knowledge is the greatest weapon in the fight against DM. Related to this currently Ethiopia government has design strategy to increase the community awareness and knowledge towards diabetes mellitus through public media and also incorporate NCD prevention and control program as one core component of health extension program to the nation [Workie et al. The health extension program in Ethiopia]. 

2. Provide a reference for the Ethiopian Government’s NCD strategy in para 2

Response: We accept your comments and appropriate reference has been provided in the revised manuscript. 

3. The consistency might be due to the widowed individuals are at risk of psychological distress or may be older?. This is too vague – what is the evidence for this?

Response: In fact authors have been put their possible reason instead of scientific evidence. Therefore, in the revised manuscript we have deleted the previous possible justification and recommended for future researcher to investigate the reason related to this issue. 

4. The second limitation reads like a bullet point

Response: Right you are. Based on your feedback necessary correction has been taken in the revised manuscript. 

5. All in all the discussion needs to more directly address the HBM and how the results might inform what interventions are needed.

Response: we thanks so much. Based on your suggestion maximum effort has been carried out to link our finding with HBM. 

6. In general the manuscript needs to be carefully checked for typos. There are a lot that need addressed.

Response: we accept your comment and appropriate correction has been made throughout the revised manuscript. 

Journal Requirements:

 Response: we accept your suggestion and the format is revised based on PLOS ONE journal style.

2. Please include additional information regarding the survey or questionnaire used in the study and ensure that you have provided sufficient details that others could replicate the analyses. For instance, if you developed a questionnaire as part of this study and it is not under a copyright more restrictive than CC-BY, please include a copy, in both the original language and English, as Supporting Information. Moreover, please include more details on how the questionnaire was pre-tested, and whether it was validated.

Response: first the questionnaire both English and Amharic version have been included as supplementary file in the revised manuscript. Second, before going to the actual data collection, the questionnaire was pretested in a similar setting outside the study district 5% (21) in Shewarobit city administration. The questionnaire was modified based on the response after it was pre-tested, and modification was made into the final version of the data collection tool. In addition, internal validity in this study was ensured through the piloting of the questionnaire in order to improve the data collection and to also identify any problems relating to understanding of the tool by respondents. With regard to external validity, the fact that the study context was specifically within the environment of urban areas, the sampling method and the sample size suggested that the findings might not necessarily be generalized to be the same in rural areas that is not similar to urban areas, because urban lifestyle differs from rural lifestyle. Moreover, Content validity of questionnaire was confirmed by 10 health education and promotion experts through calculating the content validity index (CVI) and content validity ratio (CVR).

"none"

a. Please clarify the sources of funding (financial or material support) for your study. List the grants or organizations that supported your study, including funding received from your institution.

d. If you did not receive any funding for this study, please state: “The authors received no specific funding for this work.”

Response: Authors consider your suggestion and included in the revised manuscript. 

 Response: the data availability statement has been improved. Therefore, all relevant data are within the paper and it’s supporting information files. There is no separate data set to share. 

5. Please include a copy of Table 2 which you refer to in your text on page 11.

 Response: We accept your suggestion and appropriate correction has been taken. 

Reviewers' comments:

Reviewer's Responses to Questions

Comments to the Author

1. Is the manuscript technically sound, and do the data support the conclusions?

Reviewer #1: Yes

2. Has the statistical analysis been performed appropriately and rigorously? 

Reviewer #1: I Don't Know

3. Have the authors made all data underlying the findings in their manuscript fully available?

Reviewer #1: No

4. Is the manuscript presented in an intelligible fashion and written in standard English?

Reviewer #1: No

5. Review Comments to the Author

Author’s response for reviewers 

Reviewer #1: General points:

1. The authors have not provided the ethical statement requested by the journal

Response: right you are. Based on your comment authors provided ethical statement which requested the journal in the revised manuscript. 

2. The authors need to use one of the data availability statements provided by the journal

Response: accept you suggestion and included in the revised manuscript.

3. The limitations section appears incomplete and contains what appears to be drafting note

Response: right you are. However, necessary correction has been made in the revised manuscript. 

4. Insufficient care to distinguish types of DM – it is not a single condition

Response: It is obvious that the study is done to assess public knowledge and perception towards diabetes mellitus. Therefore, the study is emphasized on both type of diabetes mellitus.

Introduction

1. Change “which means” to “equating to” when describing deaths per day.

Response: we accept your suggestion and appropriate measure has been taken in the revised manuscript. 

2. Remove erroneous ‘g’ and rephrase sentence – “Numerous studies have revealed the proportion of good knowledge respecting g diabetes mellitus was reported as…”

Response: right you are. After removing erroneous “g” we have rephrase the statement as you suggested. 

3. I would like to see the authors engage critically with the health belief model and justify its use.

Response: to the best of our effort attempts have been made extensively to link the relationship between health belief model and the aim of the present study. The aim of health belief model is increasing the perception of individuals about a health threat and directs their behaviours towards health. This model is a comprehensive pattern which represents the association between beliefs and behaviours; it has been widely applied to various health behaviours, especially screening behaviours and plays a significant role in prevention of the disease. Therefore, understanding individuals’ view points and beliefs is essential for developing the strategies of controlling diabetes. 

4. Use “scant” not “scanty”

Response: editor also raised an issue related to this “scanty” not a very scientific term. Therefore, by taking in to consideration editor and reviewer feedback, authors made necessary correction within the revised manuscript. 

Methods

1. Check the grammar and punctuation carefully – there are missing full stops, shifts of tense and duplicated words eg:

o “The town of North Shoa Zone The town”

o “The total population were 108,825, within 25,308 households” [should be ‘was’]

o “All adults living in Debre Berhan town for at least for six months…”

Response: right you are. Based on your comments all the above issue has been resolved in the revised manuscript appropriately. 

2. The authors should share the questionnaire used as an appendix or supplementary file.

Response: we accept your comment and based on your suggestion the tool has been included as supplementary file both the Amharic and English version questioners. 

3. This sentence needs to be more precise and should be referenced: “Using previous studies, as a baseline we have categorized in to agree, neutral and disagree”

Response: appropriate citation has been provided in the revised manuscript. 

Results

1. In table 1, income currency should be specified

Response: right you are. We are specified as Ethiopian birr (ETB).

2. General checking of tenses required e.g. “was widowed” should be “were widowed”

Response: We accept your suggestion. Based on your recommendation extensive grammar edition has been carried out. 

3. Table reporting knowledge of DM is labelled table 1, but it is the second table in the manuscript. The formatting of the table is also hard to read- we need additional spaces and or bold text between the sections relating to description of DM, risk factors, symptoms, and control

Response: Accept your recommendation and modification has been made in the revised manuscript. 

Discussion

1. The discussion compares findings of the study to other relevant studies well, but it does not consider the meaning of the findings presented in the paper- what does this new study add to our knowledge of diabetes in this setting?

Response: based on your comments appropriate correction has been taken to the revised manuscript.

2. Also, what do the findings suggest should happen or be prioritised? E.g. for those patients who experienced barriers to diabetes screening, what can/should be done?

Response: your concern is very interesting. Based on your comments authors have integrate perceived barrier to diabetes screening and possible recommended measures. This finding suggest that there is a need for education campaigns in the district to reduce barriers to diabetes screening about all aspects of barriers such as time consuming, and misinformation. It is therefore important to identify interventions that decrease people's perceived barriers despite their levels of knowledge on diabetes mellitus.

3. The final paragraph of the discussion is quite confusing and is not clearly written.

Response: right you are. When we reevaluate the statement lacks meaning, therefore authors totally remove the sentence and some of the idea was included in the conclusion section of revised document. 

4. The limitations paragraph is incomplete and contains drafting notes.

Response: right you are. Based on your and editor feedback necessary modification have been carried out. 

6. PLOS authors have the option to publish the peer review history of their article (what does this mean?). If published, this will include your full peer review and any attached files.

Do you want your identity to be public for this peer review? For information about this choice, including consent withdrawal, please see our Privacy Policy.

Reviewer #1: No

---

## [Editor Report · Decision Letter 1]

29 Sep 2020

PONE-D-20-04491R1

Assessment of Knowledge and Perception towards Diabetes Mellitus and Its Associated Factors among People in Debre Berhan Town, Northeast Ethiopia:

PLOS ONE

Dear Mr Shiferaw,

Thank you for submitting your manuscript to PLOS ONE. After careful consideration, we feel that it has merit but does not fully meet PLOS ONE’s publication criteria as it currently stands. Therefore, we invite you to submit a revised version of the manuscript that addresses the points raised during the review process.

We look forward to receiving your revised manuscript.

Kind regards,

Professor Kwasi Torpey, MD PhD MPH

Academic Editor

PLOS ONE

Additional Editor Comments (if provided):

The manuscript has addressed the comments provided and the quality has significantly improved. However, a number of issues remain

1. The manuscript has a lot of grammatical errors and typos that need to be a addressed. Thorough copyediting by native speaker is required. Some examples of error observed

a. Abstract: Result : Diabetes is a seriousness disease: Should read diabetes is a serious disease

bAbstract: Results : agreed in the perceived benefits should read agreed to the perceived benefits

c. Abstract: Conclusion: Behavior change has importance in prevention...... should read behavior change is important.......

d. diabetic mellitus should ready diabetes mellitus

e. Results: sociodemographic : ample should read sample

f. thorough copyedit document

2. All references need to properly checked and ensure the names and initials are accurate. They are several errors in the initials. Make sure the referencing is consistent with the journal requirement. Remove CAPS in the references (#33 and 42). Ensure first letter in names are appropriately capitalized. Include year in all the listed publications (#39). Correct all errors in references esp #2,3,4,9,13,15,17,18,19,20,23,26,30,33,42,37,39

---

## [Author Response · Author response to Decision Letter 1]

30 Sep 2020

Author’s response to editor and reviewers 

Title: Assessment of Knowledge and Perception towards Diabetes Mellitus and Its Associated Factors among People in Debre Berhan Town, Northeast Ethiopia. 

Version 2: September 30, 2020

Dear editor 

To PLOS ONE, we are pleased to resubmit for publication version of “Assessment of Knowledge and Perception towards Diabetes Mellitus and Its Associated Factors among People in Debre Berhan Town, Northeast Ethiopia” with manuscript ID of “PONE-D-20-04491” for a review as original research in PLOS ONE. The comment of the editor was highly insightful and enabled us to greatly improve the quality of our manuscript. Therefore, based on the editor concerns we have made extensive edition in our manuscript. The formatting of the text and document (text sizes and grammatical errors) were also edited. In the following pages, we have addressed yours’ concerns in a point by point format. 

 We look forward to hearing from you at your earliest convenience. 

Once again, thank you for considering our manuscript in PLOS ONE! 

Kind regards,

Wondimeneh Shibabaw Shiferaw 

On behalf of all the authors 

Author’s response to Editor Comments

The manuscript has addressed the comments provided and the quality has significantly improved. However, a number of issue remain 

1. The manuscript has a lot of grammatical errors and typos that need to be a addressed. Thorough copyediting by native speaker is required. Some example of error observed.

a. Abstract: Result: diabetes is a seriousness disease. Should read diabetes is a serious disease 

Response: We thanks and accept your suggestion. 

b. Abstract: results: agreed in the perceived benefits should read agreed to the perceived benefits

Response: Accept your suggestion 

c. Abstract: Conclusion: behavior change has importance in prevention….should read behavior change is important. ..

Response: We accept your suggestion 

d. Diabetic mellitus should ready diabetes mellitus 

Response: We accept your suggestion 

e. Results: sociodemographic: ample should read sample 

Response: We thanks and accept your suggestion. 

f. Thorough copyedit document 

Response: Based on your suggestion, we have made extensive edition in the entire document by native speaker (Dr. Ryan Bell). 

2. All references need to be properly checked and ensure the names and initials are accurate. They are several errors in the initials. Make sure the referencing is consistent with the journal requirement. Remove CAPS in the references (#33 and 42). Ensure first letter in names are appropriately capitalized. Include year in all the listed publications (#39). Correct all errors in the references esp # 2,3,4,9,13,15,17,18,19,20,23,26,30,33,42,37,39.

Response: Really the comment is constructive. Based on your feedback appropriate modification has been made in the entire references section in the revised manuscript.

---

## [Editor Report · Decision Letter 2]

5 Oct 2020

Assessment of Knowledge and Perception towards Diabetes Mellitus and Its Associated Factors among People in Debre Berhan Town, Northeast Ethiopia:

PONE-D-20-04491R2

Dear Mr Shiferaw,

We’re pleased to inform you that your manuscript has been judged scientifically suitable for publication and will be formally accepted for publication once it meets all outstanding technical requirements.

Kind regards,

Professor Kwasi Torpey, MD PhD MPH

Academic Editor

PLOS ONE
---

## [Editor Report · Acceptance letter]

7 Oct 2020

PONE-D-20-04491R2 

Assessment of Knowledge and Perceptions towards Diabetes Mellitus and Its Associated Factors among People in Debre Berhan Town, Northeast Ethiopia 

Dear Dr. Shiferaw:

I'm pleased to inform you that your manuscript has been deemed suitable for publication in PLOS ONE. Congratulations! Your manuscript is now with our production department. 

Kind regards, 

on behalf of

Professor Kwasi Torpey 

Academic Editor

PLOS ONE